# Kinematic Optimization for the Design of a Collaborative Robot End-Effector for Tele-Echography [†]

**Alessandro Filippeschi** [1,2,*] , **Pietro Griffa** [3] **and Carlo Alberto Avizzano** [1,2]

1 TeCIP Institute, Scuola Superiore Sant'Anna, Via Moruzzi 1, 56124 Pisa, Italy; c.avizzano@santannapisa.it
2 Department of Excellence in Robotics and AI, Scuola Superiore Sant'Anna, 56127 Pisa, Italy
3 ETH Zurich, Rämistrasse 101, 8092 Zürich, Switzerland; griffap@student.ethz.ch
* Correspondence: a.filippeschi@santannapisa.it; Tel.: +39-0508-82307
† This paper is an extended version of our paper published in Griffa, P.; Filippeschi, A.; Avizzano, C.A. Kinematic Optimization for the Design of a UR5 Robot End-Effector for Cardiac Tele-Ultrasonography. In Proceedings of The 3rd International Conference of IFToMM Italy, held online, 9–11 September 2020.

**Abstract:** Tele-examination based on robotic technologies is a promising solution to solve the current worsening shortage of physicians. Echocardiography is among the examinations that would benefit more from robotic solutions. However, most of the state-of-the-art solutions are based on the development of specific robotic arms, instead of exploiting COTS (commercial-off-the-shelf) arms to reduce costs and make such systems affordable. In this paper, we address this problem by studying the design of an end-effector for tele-echography to be mounted on two popular and low-cost collaborative robots, i.e., the Universal Robot UR5, and the Franka Emika Panda. In the case of the UR5 robot, we investigate the possibility of adding a seventh rotational degree of freedom. The design is obtained by kinematic optimization, in which a manipulability measure is an objective function. The optimization domain includes the position of the patient with regards to the robot base and the pose of the end-effector frame. Constraints include the full coverage of the examination area, the possibility to orient the probe correctly, have the base of the robot far enough from the patient's head, and a suitable distance from singularities. The results show that adding a degree of freedom improves manipulability by 65% and that adding a custom-designed actuated joint is better than adopting a native seven-degrees-freedom robot.

**Keywords:** design synthesis; kinematic optimization; telemedicine; human robot interaction



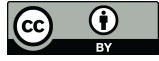

## 1. Introduction

The aging of the population makes the need for medical examinations increase every year. The available specialists are insufficient to meet this need and this shortage will worsen in the forthcoming years. Tele-medicine is a viable solution to cope with this trend and to serve areas far from hospitals.

Tele-medicine services available or under development in many of the WHO (World Health Organization) countries are typically focused on sharing examination results among specialists such as in the case of tele-radiology, tele-pathology, tele-dermatology, and tele-psychiatry [1]. However, advances in robotic and computer graphics technologies fostered the development of robotic telemedicine systems. Examples include endoscopy [2], ultrasonography [3,4] and palpation [5–7]. Among these examinations, ultrasonography (USG) is one of the most important to make a decision on a patient's need to be directed to a specialist.

In the early robotic systems designed for telemedicine (e.g., [3,8]), robotic arms were designed on purpose to place the USG probe on the patient's body. In recent years, the availability of affordable robotic arms, e.g., from Universal Robots (Energivej 25 DK-5260, Odense, Denmark) and Franka Emika (Infanteriestraße 19, 80797, Munich, Germany), has enabled the possibility to drastically reduce the costs of such systems. The most

used commercial robotic arm for tele-USG is the UR5 from Universal Robots, which has 6 DoFs (degrees of freedom). However, to the authors' knowledge, none of the proposed systems have investigated the possibility to add a degree of freedom to ease the remote manipulation of the probe.

In this paper, we study the advantages of adding such a DoF either to an end-effector to be mounted on the UR5 robot or by considering a COTS 7-DoFs arm such as the Panda by Franka Emika. We define a force manipulability metric based on the USG task and, based on this metric, we optimize the design of the end-effector in three cases: first, the end-effector is mounted on the UR5 robot and it has no DoFs with regard to the robotic arm's tip; second, one DoF is add by a rotational joint whose axis is perpendicular to the probe axis; and, third, the end-effector is mounted on the Panda robot with no additional DoFs. This work follows a preliminary study presented in [9], in which the general Yoshikawa manipulability index was adopted and the Franka Emika Panda robot was not considered. Concerning this work, we adopt an optimization metric more specific to the task, obtaining significantly different results.

The paper firstly introduces USG and current approaches to tele-USG. In Section 3, after a definition of the requirements, a model of the patient is defined and the robotic arms along with the probe are presented. This is followed by a discussion of the performance metrics that could be adopted in the design of the end-effector. The same section reports the target metric, the constraints, and the formulation of the optimization problem. Section 4 reports the details of the implementation of the problem. Section 5 reports the results of the study and their discussion. Section 6 concludes the paper.

## 2. Background

### 2.1. Ultrasonography

Ultrasonography is a technique in which sound waves are sent towards the human body, whose tissues reflect them and whose echoes are used to make a picture, called sonogram. In ultrasonography, the sonographer holds the probe of the USG machine and places it on the patient's body. This probe emits the ultrasound waves and records the reflected waves. The resulting signal is sent to a machine which reconstructs the sonogram.

The successful reconstruction of the sonogram depends on the correct positioning of the probe. The sonographer must find the correct window where the ultrasound beam is most effective to reconstruct the target and the correct orientation(s) to have the meaningful images to formulate a diagnosis. Ultrasonography is applied to several medical examinations, including those targeting the heart (echocardiography), on which this paper focuses. In echocardiography (ECG), the sonographer places the probe on the user's chest at five anatomical locations to obtain the five standard windows, i.e., suprasternal, left parasternal, right parasternal, apical, and subcostal [10] (see Figure 1).

During the examination, the sonographer places the probe on one of these locations and applies a wrench that makes the probe pin about the sought contact point until obtaining the desired acoustic window. Then, the sonographer records one or more ultrasound images and moves the probe to the next location.

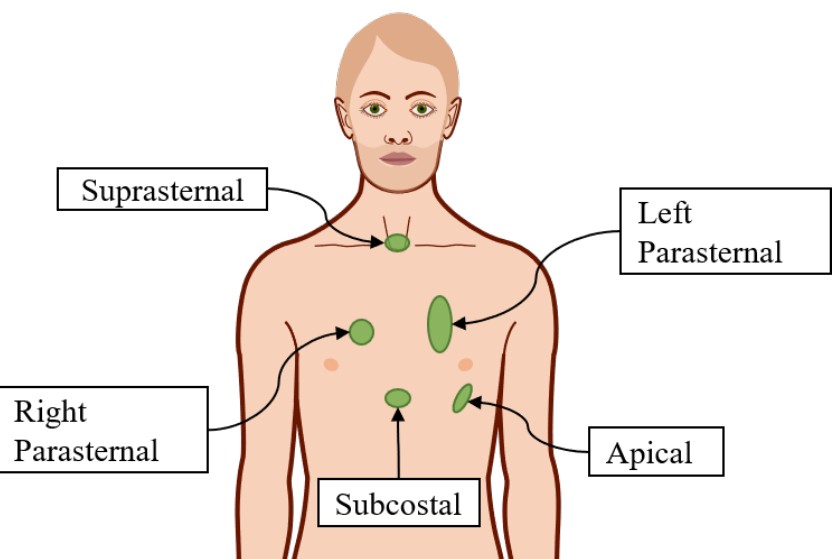

**Figure 1.** Five locations to obtain the five windows of echocardiography.

*2.2. Tele-USG Systems*

Several systems for Tele-USG have been developed in the last two decades. Some of them allow the specialist to fully control the pose of the probe at the patient site, others feature a mechanical rig that places the probe correctly on the patient torso, whereas the specialist remotely controls the orientation of the probe. A review on the Tele-USG systems was recently presented by Adams et al. [11]. In the following, we recall the main projects that have been developed throughout the years, with a final focus on those that use a robotic arm at the patient site to enable full control of the pose of the probe. An example of tele-USG system which features force feedback is described in [12,13]. The system includes a 6 DoFs robot composed of an orientable pantograph and an end-effector that allows for 3D positioning of the probe and a reasonable decoupling of translational and rotational DoFs. More recently, a complete tele-USG system was developed within the European project OTELO [14]. The system includes a 6 DoFs robot at the patient's site and a 6 DoFs haptic interface at the expert site. The robot at the patient site is custom-designed and has PPRRRP kinematics, with the three R joints in a wrist configuration. In this line, Arbeille et al. [3] developed a tele-USG system which works over a satellite link to make available echography examination for astronauts (TERESA project [15]). In their system, the patient and the expert sites are linked by a videoconference system. At the patient's site, a non-specialist operator places the robot (ESTELE) on the patient. This robot, purposely designed for USG, is composed of a rigid structure that is placed on the patient and that hosts an RRR spherical wrist manipulator that orients the probe on the patient's body. The following ARTIS project (European Space Agency contract number ESA No. 21210/07/NL/HE) further developed this concept, resulting in a simpler robot (the kinematic optimization is reported in [16]) at the patient site, which still needs to be held by an assistant. In recent years, the MELODY system [17] stemmed from the TERESA and ARTIS projects and has been commercialized by AdEchoTech. In this latter system, the robot at the patient site is held by a passive manipulator that balances the device's weight and helps the assistant to place the robot on the patient's torso. A similar approach has been adopted for the FASTele system [18], which aims at a prompt intervention during an emergency. In addition, in this case, the robot at the patient site allows for partial control of the probe. In particular, the robot has PRRP kinematics where the joint axis is parallel to the patient's longitudinal axis, the two R joints determine the probe's main axis orientation, and the latter P joint is directed along the probe's axis and features two springs to keep the contact with the patient's body. Differently from the OTELO system, this robot allows only for small displacements of the probe, whereas the gross motion

from one anatomical location to the other is carried out by a non-specialist, under video surveillance of the expert.

In the TER tele-USG system [19], a different slave robot at the patient site is proposed. This robot decouples the gross positioning of the probe from its pose fine refinement on the patient's body. This robot has a kinematic structure composed of two parts: The first is parallel, with a ring that has a planar motion and that is actuated by either four McKibben artificial muscles or four DC motors trough four belts in an antagonistic configuration. The second is serial and composed of a wrist and a prismatic joint for fine-tuning of the probe position along its axis.

In the ReMeDi tele-examination system [8], a 6 DoFs serial arm is mounted on a mobile base to allow the robot to achieve a correct position around the patient and to allow the sonogrpaher to place the probe correctly from remote using a multimodal diagnostician user interface [4].

This latter system, as well as the many recent tele-USG systems, adopts a robotic arm with serial kinematics at the patient's site. Moreover, most of these recent systems use commercial robotic arms to improve the feasibility and to make both the development of such systems and the final product more affordable. For example, the system proposed in [20] uses a Viper s650 arm, whereas the system developed by Mathur et al. uses a KUKA LWR arm [21]. The most used commercial arm is thus far the Universal Robot UR5 [22–25]. The main reasons are the compliance with the ISO 10218-1:2006, which makes it usable as a collaborative robot, and its affordability. At the same time, its kinematic structure makes its workspace cover almost all the patient's torso. The Panda arm from Franka Emika has similar advantages. It has 7 DoFs, but it has a smaller workspace and lower joint torques limits with regard to the UR5 arm. Tele-USG setups the uses the Panda arm include the works presented by Sandoval et al. [26] and Kaminski et al. [27]. In the former, the probe is attached to the robot's flange using a compliant prismatic joint with variable stiffness for the patient's safety, whereas, in the latter, specifically aimed at Thyroid USG, the probe is rigidly attached to the end effector of the arm.

Thanks to the aforementioned advantages, most of these systems use a rigid end effector, which is attached to the last link of the arm, to hold the probe. None of these systems investigated the possibility to add one actuated DoF to the end-effector to optimize the dexterity of the robot at the patient's site in all the positions required for echocardiography. In particular, we study the effect of adding one DoF to the end-effector of the UR5 robot. Moreover, we compare this solution to the adoption of the Panda arm, which has natively 7 DoFs, is cost-effective, and brings advantages similar to the UR5 robot.

## 3. End-Effector Design Optimization

### 3.1. Requirements

In the USG examination task, the robot at the patient site must reach any target point on the torso and guarantee to move without collisions between any two anatomical locations. Therefore, the workspace of the robot has to include the torso. Moreover, the robot has to allow the sonographer to change the probe's orientation while exerting the contact wrench. According to experimental measurements acquired within the ReMeDi project, the main component of the contact force is along the probe long axis and equals $12 \pm 3$ N, whereas the remaining two components are approximately equal to $5 \pm 1$ N. The torque component along the probe long axis is approximately 0.02 Nm, whereas the remaining two components are equal to $0.5 \pm 0.3$ Nm. In addition to the capability to maintain the contact wrench, it is important to guarantee that the robot's configurations are far from singularities when the probe moves from one location to another over the patient. For the safety and the acceptability of the system, it is indeed not desirable that the manipulator makes weird and fast movement when leaving one contact point to reach the next. Finally, we treat the problem as quasi-static, focusing only on the contact phases. In fact, the time spent in any anatomical location is much higher than the time needed to

move from one location to another. Therefore, it is not worth maximizing the speed of the end-effector between two locations.

Model of the Patient and Target of the ECG

The definition of a patient's model is a difficult task for two main reasons: first, patients can have very differently shaped and sized bodies. Second, patients can lie on the back or on the side. In this paper, we make some simplifications to make the problem tractable. First, we adopt an average-sized chest, and we approximate its shape to an elliptic cylinder aligned to the human longitudinal axis, whose semi-axis lengths are 26 and 15 cm, respectively. Second, we assume that the patient lies on their back, which is largely the most common case in ECG. Third, we assume the heart as a particle target $H$ placed inside the chest. Fourth, we define five target points $E_i, i = 1, \ldots 5$ on the chest to represent the probe positions where the standard windows are sought (see Section 2.1). Finally, we sample the chest using 16 points $P_i, i = 1, \ldots, 16$ equally distributed in the longitudinal and lateral directions in the coronal plane, and we extend the samples set by adding points $E_i$ to $P_i$, thus having a total of 21 points. Figure 2 shows the model of the chest along with the heart, the $E_i$ targets, and the sample points $P_i$ along with the reference frame $\Sigma_s$ used to define the pose of the patient. The origin of $\Sigma_s$ is $S$, which is at the base of the throat, the $y_S$ axis is directed towards the head in the longitudinal direction, whereas the $z_S$ axis points upwards. Points $E_i$ and $P_i$ are defined in $\Sigma_S$.

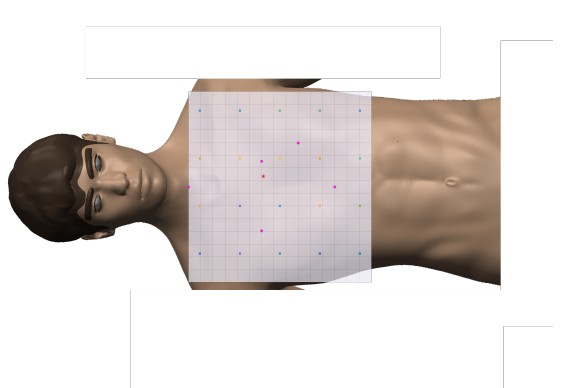

(**a**) Patient and model of her/his chest

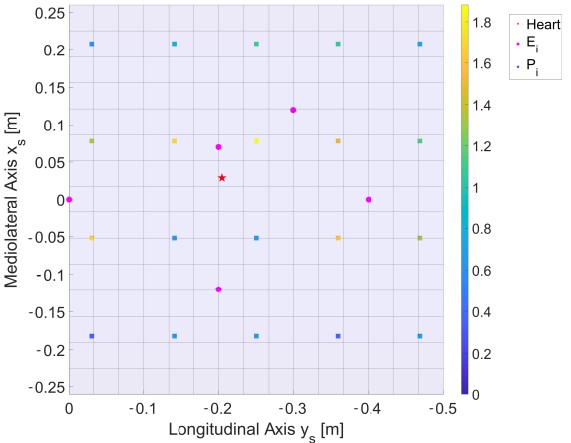

(**b**) Coronal view of the chest model along with heart, target points $E_i$ and sample points $P_i$, whose color represent the respective $\lambda_i$ according to the colorbar

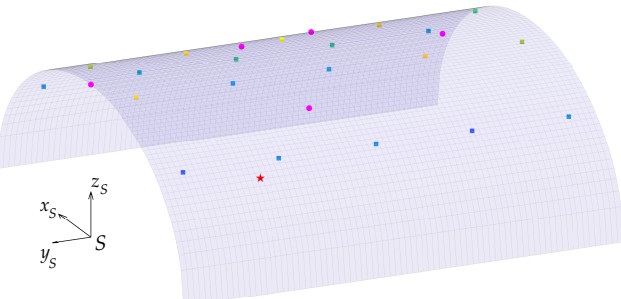

(**c**) 3D view of the model of the chest along with $\Sigma_S$ frame.

**Figure 2.** The mathematical model of the patient. The heart is represented as a red asterisk, whereas target points $E_i$ are plotted in orange and the sample points $P_i$ in green.

Based on the aforementioned definitions, the target of the ECG is defined as placing the USG probe on the patient in every point $P_i$ while pointing the axis of the probe towards the heart $H$. This task sets five of the six DoFs of the probe. In fact, the rotation of the probe about its axis remains free. However, to use the arm in a teleoperation setting, the doctor needs to have full control of the pose of the probe. Therefore, in Section 3.6, the orientation of the lateral axis of the probe will also be set to replicate commonly used orientations of the probe.

### 3.2. UR5 and Panda Kinematics

This section introduces the kinematic models of the three adopted arms. The Universal Robots UR5 is a 6 DoFs manipulator which includes revolute joints only. Its kinematics is similar to an anthropomorphic arm, with the noticeable difference that the last three R joints are not arranged in a spherical wrist fashion, so that all six joints contribute to both the translational and rotational motion of the end-effector [28]. A description of the kinematics of the robot is reported in Figure 3a, whereas Table 1 reports the Denavit–Hartenberg parameterization of the robot. In Figure 3a, $\Sigma_6$ and $\Sigma_7$ are the frames attached to the end-effector without and with additional joint, respectively.

The Franka Emika Panda is a 7 DoFs manipulator composed of eight links connected by revolute joints. Its kinematics includes a full shoulder, an elbow, and a wrist, thus being similar to a human arm. As for the UR5, the wrist is not spherical, hence all joints contribute to both the position and orientation of the end-effector. The robot is shown in Figure 3, whereas Table 1 reports the Denavit–Hartenberg parameters as reported by the manufacturer. For the Panda robot, the same notation of the UR5 with additional DoF holds.

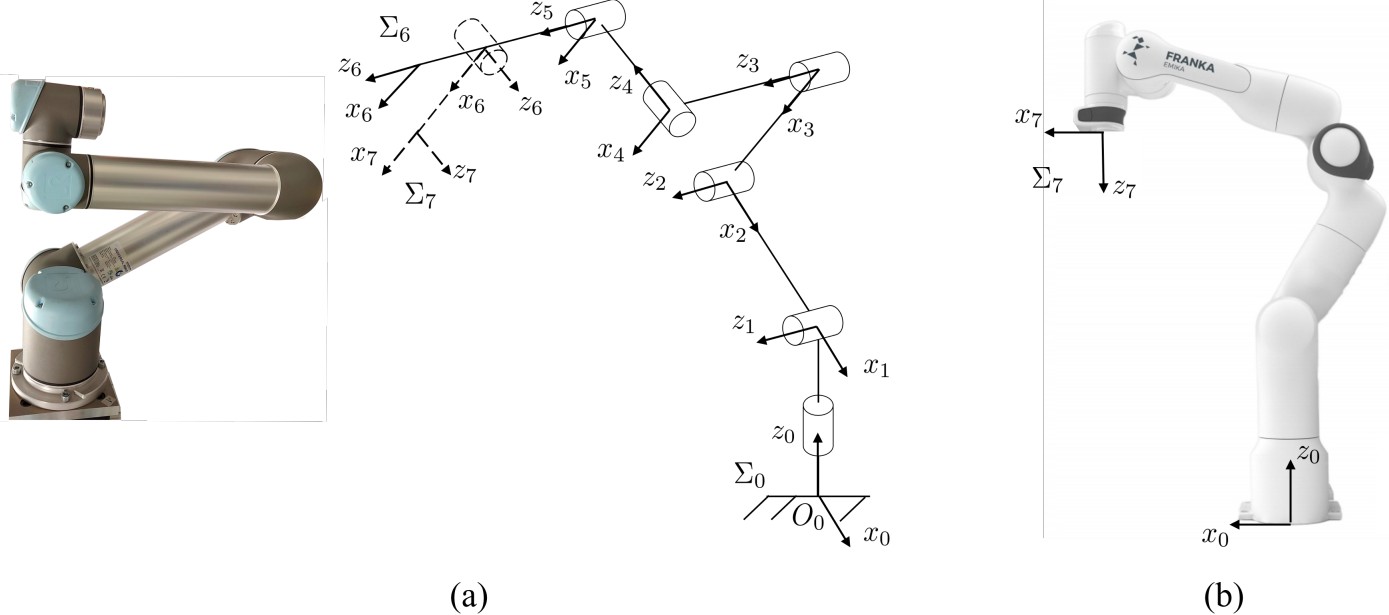

**Figure 3. (a)** The Universal Robot UR5 robot and the adopted kinematic model without and with additional joint. (**b**) The Panda robot from Franka Emika along with the first and the last frames, $\Sigma_0$ and $\Sigma_7$, respectively.

$\mathbf{q} = [q_1, \ldots q_n]^T$ and $\dot{\mathbf{q}} = [\dot{q}_1, \ldots \dot{q}_n]^T$ are the joint angles vector and the vector of their time derivatives, respectively. The geometric Jacobians $J$ of the three manipulators satisfies:

$$\begin{bmatrix} \mathbf{v}_C \\ \omega_p \end{bmatrix} = J\dot{\mathbf{q}} \tag{1}$$

where $\mathbf{v}_C$ is the velocity of the end-effector, i.e., the center of the surface of the probe that goes in contact with the patient's skin; $\omega$ is the angular velocity of the end-effector; $n = 6$ for the UR5 robot; and $n = 7$ for the remaining two arms.

**Table 1.** Denavit–Hartenberg parameters for the definition of the kinematics of the two robots. The seventh frame of the UR5 with additional joint is drawn as separate from the sixth for clarity of representation, even though their origins are superimposed. * UR5 without additional joint; ** UR5 with an additional joint; $d_6$ is a design parameter which is part of the optimization.

| UR5 | | | | |
|---|---|---|---|---|
| link | $a_i$ | $\alpha_i$ | $d_i$ | $\theta_i$ |
| 1 | 0 | $\pi/2$ | 0.0895 | $\theta_1$ |
| 2 | −0.4250 | 0 | 0 | $\theta_2$ |
| 3 | −0.3922 | 0 | 0 | $\theta_3$ |
| 4 | 0 | $\pi/2$ | 0.1091 | $\theta_4$ |
| 5 | 0 | $-\pi/2$ | 0.0946 | $\theta_5$ |
| 6 * | 0 | 0 | 0.0823 | $\theta_6$ |
| 6 ** | 0 | $-\pi/2$ | $d_6$ | $\theta_6$ |
| 7 | 0 | 0 | 0 | $\theta_7$ |
| **Panda** | | | | |
| link | $a_i$ | $\alpha_i$ | $d_i$ | $\theta_i$ |
| 1 | 0 | 0 | 0.333 | $\theta_1$ |
| 2 | 0 | $-\pi/2$ | 0 | $\theta_2$ |
| 3 | 0 | $\pi/2$ | 0.316 | $\theta_3$ |
| 4 | 0.0825 | $\pi/2$ | 0 | $\theta_4$ |
| 5 | −0.0825 | $-\pi/2$ | 0.384 | $\theta_5$ |
| 6 | 0 | $\pi/2$ | 0 | $\theta_6$ |
| 7 | 0.088 | $\pi/2$ | 0.107 | $\theta_7$ |

### 3.3. End-Effector Kinematics

Echocardiography probes (see Figure 4) have a longitudinal axis $z_p$ in their prominent direction and a second perpendicular axis $x_p$ that defines the ultrasound plane $\pi_p$ (we do not explicitly consider the case of 3D ECG, for which a similar treatment could be proposed). ECG probes are typically symmetric with regard to $\pi_p$ and to a plane perpendicular to $\pi_p$ which passes through $z_p$. The image obtained by ultrasound is a circular sector $\Gamma$ which lies in $\pi_p$ and is symmetric with regard to $z_p$. In the recent ECG probes, the $x_p$ axis can be rotated via software. Therefore, it is defined in the software reference configuration, in which the rotation of the ultrasound plane with regard to $z_p$ is 0. This typically means that $x_p$ lies in one of the two symmetry planes of the probe. Additionally, we define two points , $C$ and $G$ on the $z_p$ axis (see Figure 4). The former is the centroid of the contact surface of the probe with the patient. The latter is the point at which the probe is rigidly attached to the end-effector.

The centroid $C$ is assumed to lie on the $z_p$ axis. Given the symmetry planes of the probe and that the image $\Gamma$ is symmetric with regard to $z_p$, it is reasonable to assume that the sonographer operates to have $C \in z_p$ in order to have the best image. Moreover, from the requirements, we have forces in the order of 5N in the $x_p$ and $y_p$ directions, whereas the torque along $z_p$ is nearly 0. Therefore, $C$ cannot lie far from $z_p$. We also assume that $\left\| \overrightarrow{GC} \right\| = 4$ cm, which can be easily obtained by design for the most common ECG probes.

The point $G$ lies on $z_p$ by the design of the end-effector. A different choice would produce unnecessary torques due to $\mathbf{w}$, which the attachment to the-effector would have to balance. This design constraint does not reduce the generality of the approach. In fact, point $G$ can be arbitrarily placed in the frame of the end-effector, and we define $\mathbf{p}_G = [x_G y_G z_G]^T$ the position of $G$ in the end-effector frame $\Sigma_n$.

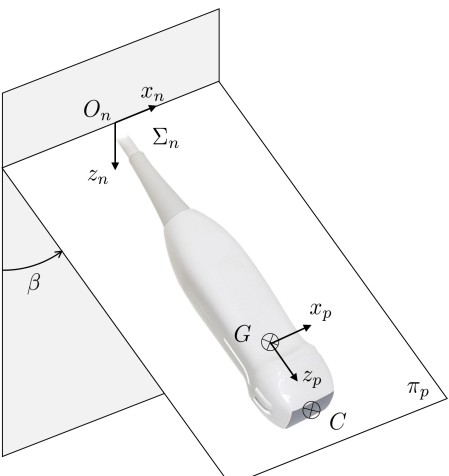

**Figure 4.** Echocardiographic probe along with the definition of its kinematic variables.

Whereas 7-DoFs manipulators offer a kinematic redundancy that can be exploited to optimize the orientation of the probe with joints in the proximity of the patient, the UR5 does not have this possibility. Therefore, in absence of a seventh DoF, we decided to add a further design variable $\beta$, which is the fixed angle by which frame $\Sigma_6$ is rotated about $x_6$ to obtain the final orientation of the probe (see Figure 4). The pose of the probe with regard to $\Sigma_n$ is then obtained by means of the homogeneous transformation matrix

$$T_p = \begin{bmatrix} 1 & 0 & 0 & x_G \\ 0 & \cos\beta & -\sin\beta & y_G \\ 0 & \sin\beta & \cos\beta & z_G \\ 0 & 0 & 0 & 1 \end{bmatrix} \tag{2}$$

where $\beta$ is a design variable for the UR5 without additional DoF, whereas it is used in the other two cases to properly orient $z_p$, i.e., $\beta = \pi/2$ for the UR5 with additional DoF, and $\beta = 0$ for the Panda robot.

### 3.4. Optimization Variables

The design of the end-effector includes four main variables: the position of $G$ with regard to the terminal link of the robot, and the inclination of the plane $\pi_p$ with regard to the terminal link $x_n$ axis, i.e., the fixed angle $\beta$ for the UR5 without additional joint.

In addition to these variables, we include the position of the patient with respect to the robot base in the optimization problem. To do that, we consider that the patient is steady with respect to the global frame $\Sigma_0$ attached to the base of the robot (see Figure 3) and that the pose of its reference frame $\Sigma_s$ with regard to $\Sigma_0$ is defined by

$$T_s = \begin{bmatrix} \mathbf{I} & \mathbf{p}_s \\ \mathbf{0} & 1 \end{bmatrix} \tag{3}$$

where $\mathbf{p}_s = [x_s y_s z_s]^T$. Therefore, when there is no additive DoF in the UR5 robot, the seven optimization variables are contained in the array $\mathbf{x}_A = [x_s y_s z_s x_G y_G z_G \beta]$. In the other two cases, the optimization variables included in the array $\mathbf{x}_B$ are six: $\mathbf{x}_B = [x_s y_s z_s x_G y_G z_G]$. To simplify the notation, when possible, we refer to $\mathbf{x}$ to mean the optimization array.

### 3.5. Objective Function

The experimental results reported in Section 3.1 set a target for the tele-USG task that will be exploited to define the objective function of the design optimization. The main objective of the design is overall good manipulability of each arm, especially in the areas that surround the target points $E_i$.

In the literature, several manipulability measures have been proposed to compare different manipulators and to optimize their design [29,30]. In particular, when the capability of exerting wrenches is concerned, Patel et al. [29] moved from the classical Yoshikawa's analysis based on the calculation of the force ellipsoids to define task-specific indices. These indices take into account the wrench $\mathbf{w}$ that the arm has to produce at the end-effector.

In the simplest case in which no specific cost hierarchy is to be applied to the components of the manipulator torque vector $\tau$ and all the components of the wrench $\mathbf{w}$ are equally important, the Yoshikawa manipulability index provides a well-recognized measure of local manipulability, which can be easily extended by integration to a global index of manipulability. However, in the case in which the joint torque limits are not the same for all joints, and when there are preferable directions in the wrench space for the wrench exertion, a better choice to evaluate the manipulability was initially proposed by Bicchi et al. [31] and elaborated for the force analysis in [32]. In this latter work, the authors proposed the following metric:

$$\eta = \frac{\mathbf{w}^T W_u \mathbf{w}}{\tau^T W_\tau \tau} \tag{4}$$

where $W_u$ and $W_\tau$ are positive definite and allow for applying weights to the different components of $\mathbf{w}$ and $\tau$.

Other approaches to the optimization of manipulability are based on the wrench requirements and are task-specific indexes. These include the task-dependent performance index [33], in which the author proposed to minimize the weighted sum of the normalized differences between the transmission ratios desired for the task, i.e., the task ellipsoid semi-axis lengths, and the actual transmission ratios, i.e., the force ellipsoid semi-axis lengths. Though computationally simple, the application of the method to this problem requires careful tuning of the weights to account for the different units of the wrench components and to include the different cost of the manipulator actuators in terms of torque. Therefore, Equation (4) is selected as the metric.

To set the optimization problem as a minimization problem, the metric

$$\mu = \frac{1}{\eta} = \frac{\tau^T W_\tau \tau}{\mathbf{w}^T W_u \mathbf{w}} \tag{5}$$

is adopted and elaborated for the purpose of this study. The matrix $W_t$ is introduced to account for the different capabilities of the joint actuators. It is a diagonal matrix whose elements are the inverse of the maximum joint torques provided by the manufacturer: $W_t = \text{diag}(1/t_{m_1}, \ldots, 1/t_{m_n}), \quad i = 1 \ldots n$. In the case of the additional joint of the UR5, $t_{m_n} = t_{m_{n-1}}$. The matrix $W_u$ is diagonal as well, and it accounts for the task defined in Section 3.1: $W_u = \text{diag}(w_1, \ldots, w_6)$, where $w_1 = 12, w_2 = 5, w_3 = 5, w_4 = 0.5, w_5 = 0.5, w_6 = 0.02$ are obtained from the requirements reported in Section 3.1. This choice allows us to take in consideration the physical units both in the joint and the task space.

From the statics of the manipulator, we have

$$\tau = J^T \mathbf{w}, \tag{6}$$

Since the singularity configurations of the manipulators are avoided in the optimization process, $J$ is always full rank, with $\dim \mathcal{R}(J) \geq 6$, where $\mathcal{R}(J)$ is the range of $J$. This means that all wrenches can be obtained using active torques and that no wrench applied to the end-effector can be balanced by a null vector in the torque space. By putting Equation (6) into Equation (5), we obtain

$$\mu = \frac{\mathbf{w}^T J W_t J^T \mathbf{w}}{\mathbf{w}^T W_u \mathbf{w}} \tag{7}$$

which is a generalized Rayleigh quotient that we want to minimize. Since both $W_t$ and $W_u$ are diagonal, the minimization of $\mu$ is straightforward and reduces to

$$\tilde{\mu} = \arg\min_{\sigma} W_u^{-1} J W_t J^T \mathbf{w} = \sigma \mathbf{w} \tag{8}$$

which only requires the computation of the eigenvalues of the matrix $W_u^{-1} J W_t J^T$. The local index $\tilde{\mu}_i$ is evaluated at each patient target point $P_i$ to compute a global manipulability index

$$\mathcal{M} = \sum_{i=1}^{21} \lambda_i \tilde{\mu}_i \tag{9}$$

The mentioned

$$d_i = \arg\min_{d_j} d_j = \left\| \overrightarrow{P_i E_j} \right\| j = 1 \ldots 5, \tag{10}$$

$\tilde{E}_i$, and the corresponding target point, $\lambda_i$, are calculated as follows

$$\lambda_i = \nu_i \left( 1 - e^{\frac{-1}{20 d_i}} \right) \quad \begin{cases} \nu_i = 1.5 & \text{for } \tilde{E}_i = E_5 \\ \nu_i = 3 & \text{otherwise} \end{cases} \tag{11}$$

that implies $\lambda_i \in [0.3, 3]$.

### 3.6. Constraints

The first constraint of the problem is the accomplishment of the task while keeping the manipulator far from singularities. In the definition of the task, the rotation of the probe around $z_p$ is not left as an optimization variable, unnecessarily increasing the optimization domain. Instead, five $\tilde{x}_i$ $i = 1 \ldots 5$ orientations for the probe lateral axis $x_p$ are defined for each of the $E_i$ points so that the orientation of the plane $\pi_p$ is nearly correct when in the reference configuration. For each point $P_i$, the orientation $\tilde{x}$ of the nearest target point $E$ is selected. For each point $P_i$, an inverse kinematics problem is then solved, i.e., the vector $\mathbf{q}_i$, which defines the configuration of the manipulator is calculated by imposing that

$$\begin{aligned} C &= P_i + \mathbf{p}_S \\ z_p &\parallel \overrightarrow{P_i H} \\ x_p &\parallel \tilde{x}_i \end{aligned} \tag{12}$$

For the solution of the inverse kinematics, six joint configurations spanning the joint space within the joint limits are defined as starting points. These starting points are used to solve the inverse kinematics numerically by means of the weighted damped least-squares method described in [34]. For each $P_i$, three outcomes are possible: First, no solution is found. In this case, the optimization array $\mathbf{x}$ is penalized by setting $\tilde{\mu}_i = 10^8$, which makes it not competitive against feasible solutions. Second, one solution $\mathbf{q}_i$ is found. In this case, the condition number of the manipulator $\kappa(\mathbf{q}_i)$ is computed and compared against the acceptability threshold $\tilde{\kappa}$, which is set by approaching each manipulator to a singularity configuration. If $\kappa(\mathbf{q}_i) \geq \tilde{\kappa}$, the array $\mathbf{x}$ is penalized by setting $\mu_i = 10^8$. Finally, if more solutions are found, they are filtered by the condition number criterion. Then, they are ranked according to $\tilde{\mu}_i$ and the best one is selected to contribute to $\mathcal{M}$. This procedure does not guarantee that the best solution (in terms of $\tilde{\mu}_i$) of the inverse kinematics is found, but the proposed method was tested in several configurations to check that at least the known multiple solutions (e.g., elbow-up vs. elbow-down) of the inverse kinematics were obtained in the target points for some positions of the patient with regard to the base. A thorough exploration requires, at least for the 7 DoFs manipulators, at each optimization step, the solution of a nested optimization problem, which would make the time to find a solution to increase exponentially, without guaranteeing to find an optimal solution.

The second set of constraints (Equation (13)) is imposed for feasibility, safety, and acceptability reasons: the base cannot be placed far from the patient, because part of the

patient chest would be outside of the manipulator workspace. Moreover, the base cannot be too close to the patient's head or between her/his legs.

$$\mathbf{p}_{sm} \leq \mathbf{p}_s \leq \mathbf{p}_{sM} \tag{13}$$

The third constraint set 14 is added to limit the size of the end-effector, to avoid it being hard to control, possibly making the joint torques insufficient to obtain **w**, expensive, and cumbersome to realize, as it would be bigger and bulkier due to the higher stress.

$$\mathbf{p}_{Gm} \leq \mathbf{p}_G \leq \mathbf{p}_{GM} \tag{14}$$

A fourth constraint limits the probe orientation. This is a limit for the angle with which the probe is mounted, in the case of a rigid end-effector, and a limit on $\theta_7$ in the other case.

$$\begin{cases} \beta_m \leq \alpha \leq \beta_M & 6 \text{ DoFs} \\ \theta_{7m} \leq \theta_7 \leq \theta_{7M} & 7 \text{ DoFs} \end{cases} \tag{15}$$

Finally, it is necessary to guarantee that the robot avoids any possible collision in its motion around the patient's body. We did not include this constraint in the optimization problem, thus speeding its solution up, but we verified ex-post (see Section 4) that, for some trajectories around the patient, there were no collisions.

## 4. Implementation

The whole optimization was implemented in Matlab R2019A except for the collision avoidance that was modeled in Gazebo. Peter Corke's Robotics Toolbox [35] was used for the definition of the manipulator, the computation of the Jacobians $J$ and the solution of the inverse kinematics using the "ikine" method of the SeialLink class. The robots were created starting from the existent models of the UR5 and Panda arms as new instances of the SerialLink class. The end-effector was then implemented as a tool of the SerialLink object.

The optimization problem was implemented using the Matlab Optimization Toolbox and the MultiStart algorithm, which allows a thorough exploration of the optimization domain and the exploitation of parallel computing. The MultiStart algorithm samples uniformly the optimization domain within the bounds and runs local solvers to find local minima. After all start points are evaluated, the algorithm compares the local solvers' solution to return a "global" minimum. In our implementation, after a pilot trial to evaluate the order of magnitude of $\mathcal{M}$, we set the "FunctionTolerance" parameter, which is used to compare the minima of $\mathcal{M}$, to $10^{-7}$, and the "XTolerance" parameter, which is the minimum distance between two **x** points to be considered as separated, to $10^{-4}$. The selected solver is "fmincon", which is a gradient-based method that exploits an interior point algorithm suitably designed to account for both equality and inequality constraints. The torque limits of the UR5 arm are $t_{m1} = t_{m2} = t_{m3} = 150$ Nm and $t_{m4} = t_{m5} = t_{m6} = 28$ Nm. The torque limits of the Panda arm are $t_{m1} = t_{m2} = t_{m3} = t_{m4} = 87$ Nm and $t_{m5} = t_{m6} = t_{m7} = 12$ Nm.

Constraints defined in Equations (13)–(15) were set as bounds of the optimization array components. In particular, regarding Equation (13), we set $\mathbf{p}_{sm} = [0.35, 0.10, -0.40]$ m. The first component ensures that the base of the robot is outside the patient. The second component guarantees that the base is at least 3 cm from the neck along the patient's longitudinal axis. The latter represents the vertical displacement of the patient with regard to the base. It was set after pilot trials that showed increasing difficulties for the arm to reach all the target points when moving the patient too much below the robot. We set $\mathbf{p}_{sM} = [0.90, 0.60, 0.10]$ m. These maximum values were set after pilot trials, as the arm encountered increasing difficulties to reach all target points. Regarding the position of probe with regard to the arm flange, we limited the range in the plane of the arm terminal flange to $\pm 0.15$ m in both directions. Larger displacements would lead to a cumbersome end-effector, with a relevant effect on the magnitude of the wrench at the end-effector that the arm can balance. The limits in the direction perpendicular to the flange were set to 0.05 and 0.30 m. The former gives sufficient room for the design of the end effector (0 means

that the probe penetrates into the arm's flange), while the latter value was set after pilot trials. Finally, we set $\theta_{7m} = \beta_m = -3\pi/4$ and $\theta_{7M} = \beta_M = 3\pi/4$. These values allow the probe to point towards the arm's terminal flange, and they were retained sufficient to explore the reasonable configurations. The constraints defined in Equation (12) were embedded in the computation of the objective function (see Algorithm 1).

The optimal array $\tilde{\mathbf{x}}$ minimizes $\mathcal{M}$ while satisfying the constraints. Algorithm 1 synthesizes the whole optimization procedure to find $\tilde{\mathbf{x}}$.

---

**Algorithm 1:** Optimization algorithm

---

**Data:** sample points $P_i$, weights $\lambda_i$
**Result:** optimal
create the manipulator MN;
set $W_u$, $W_t$, $\mathcal{M} = 0$ ;
compute $P_i$, $\lambda_i$;
compute inverse kinematics starting points $\mathbf{q}_0$;
set bounds in MultiStart according to Equations (13)–(15);
**while** *optimization stop criteria in MultiStart are not met* **do**
 set $\mathcal{M} = 0$;
 get $\mathbf{x}$ from MultiStart;
 **if** *MN == UR5* **then**
  update MN tool: MN.tool($x_G, y_G, z_G, \beta$);
 **else if** *MN == UR5 + 1 DoF* **then**
  update $d_6$ in Denavit Hartenberg table: $d_6 = z_G$;
  update MN tool: MN.tool($x_G, y_G$);
 **else if** *MN == Panda* **then**
  update MN tool: MN.tool($x_G, y_G, z_G$);
 **end**
 compute $P_i + \mathbf{p}_S$;
 **for** $i \leftarrow 1$ **to** 21 **do**
  compute target pose $T_i$ according to constraints (12);
  **for** $j \leftarrow 1$ **to** 6 **do**
   $\mathbf{q}_{i,j}$ = MN.ikine($T_i$,$\mathbf{q}_{0_j}$);
   set $\tilde{\mu}_{i,j} = 10^8$;
   **if** *exist* $\mathbf{q}_{i,j}$ **then**
    **for** $k \leftarrow 1$ *to number inverse kinematics solutions* **do**
     $J_{i,j}$ = MN.jacobe($\mathbf{q}_{i,j,k}$) (compute the Jacobian);
     compute $\kappa_{i,j,k}$;
     **if** $\kappa_{i,j,k} < \tilde{\kappa}$ **then**
      compute $\sigma_{i,j,k,l} = \text{eig } W_u^{-1} J_{i,j} W_t J_{i,j}^T$;
      $\tilde{\mu}_{i,j,k} = \min_l \sigma_{i,j,k,l}$;
     **end**
    **end**
    $\tilde{\mu}_{i,j} = \min_l \sigma_{i,j,l}$;
   **end**
  **end**
  $\tilde{\mu}_i = \min_j \tilde{\mu}_{i,j}$;
  $\mathcal{M} = \mathcal{M} + \tilde{\mu}_i$;
 **end**
 return $\mathcal{M}$ to MultiStart;
**end**

---

Finally, to verify the eventual occurrence of collisions when moving around the patient's chest, and possibly avoid them, the manipulators comprehensive of all their elements (robotic arm, the box representing the end-effector, and body of the patient) were modeled in ROS-Gazebo. Simulations were run imposing the trajectories of interest among points $P_i$ and the collision among the elements were identified.

## 5. Results

The results of the optimization for the three manipulators are reported in Table 2. In addition to $\mathcal{M}$, this table reports the ratio

$$\epsilon = \max \tilde{\mu}_i / \min \tilde{\mu}_i \qquad (16)$$

that allows an evaluation of the uniformity of the arm's behavior across the patient's torso.

Feasible solutions have been found for each manipulator. None of the components of the **x** array is on the boundaries set by Equations (13)–(15), making us confident that the choice of the boundaries has not excluded optimal solutions nearby the boundaries.

The obtained optimal vectors reported in Table 2 were used to calculate the objective metric $\tilde{\mu}$ at the target points, thus enabling a detailed evaluation of the behavior of the arms. To have a richer representation of the performance of the arm, 48 target points equally spaced on the patient's torso were used. Figure 5 shows the objective metric $\mu$ evaluated at these points. With regard to the arms' performance, we report in Table 2 the ratio between the maximum and minimum value of $\mu$ calculated in these target points.

The simulations run in Gazebo showed that it is possible to reach every $P_i$ without colliding with the patient or having self collisions.

**Table 2.** Optimization results for the three manipulators. UR5+1 stands for Ur5 with additional joint.

| Robot | $x_S$ [m] | $y_S$ [m] | $z_S$ [m] | $x_G$ [m] | $y_G$ [m] | $z_G$ [m] | $\beta$ [deg] | $\mathcal{M}$ [$10^{-4}$] | $\epsilon$ |
|---|---|---|---|---|---|---|---|---|---|
| UR5 | 0.55 | −0.23 | 0.11 | 0.017 | −0.019 | 0.161 | −65.9 | 5.56 | 8.54 |
| UR5 + 1 | 0.73 | −0.33 | 0.20 | 0.037 | 0.005 | 0.123 | - | 1.97 | 3.23 |
| Panda | 0.39 | −0.17 | 0.31 | 0.023 | 0.071 | 0.146 | - | 5.28 | 14.1 |

*Discussion*

The results of the optimization, reported in Table 2, show that adding a DoF to the UR5 robot improves the performance index significantly (65%). The overall value of $\mathcal{M}$ and the smaller value of $\epsilon$ support this statement.

The same benefits are not present when using the Panda arm. This is likely due to the smaller size of the arm and the limited range of motion of the joints. These limited the feasible optimization domain, especially regarding $\mathbf{p}_s$, thus limiting the possibility to achieve the performance level of the UR5 with one additional DoF. This statement is also consistent with the large value of $\epsilon$ (see Table 2 and Figure 5c). Figure 5c shows that $\tilde{\mu}$ is relatively high along a line directed as $x_s$ is passing through the arm base and around this line in the vicinity of the base (except for a second line parallel and next to this one), whereas $\tilde{\mu}$ increases far from the arm base. A possible explanation, which is also supported by the relatively high value of $\sqrt{x_G^2 + y_G^2}$, is that the optimization process was forced to keep the base close to the patient to guarantee that all targets could be reached. This makes the arm assume very compact configurations in the proximity of the base, with detrimental effects on the manipulability even in the presence of kinematic redundancy.

Differently from the Panda, the UR5 has a large workspace, which covers the patients' torso with a good margin. However, it is not redundant. The large size and a larger optimization domain (seven variables instead of six) made $\epsilon$ much smaller than for the Panda. Figure 5a shows that, apart for one target point, $\tilde{\mu}$ is pretty uniform. If that point is removed, the ratio drops to 4.82, which is much closer to the case of UR5 with an additional joint than to the case of Panda arm. Curiously, the overall performance $\mathcal{M}$ is similar to the Panda arm.

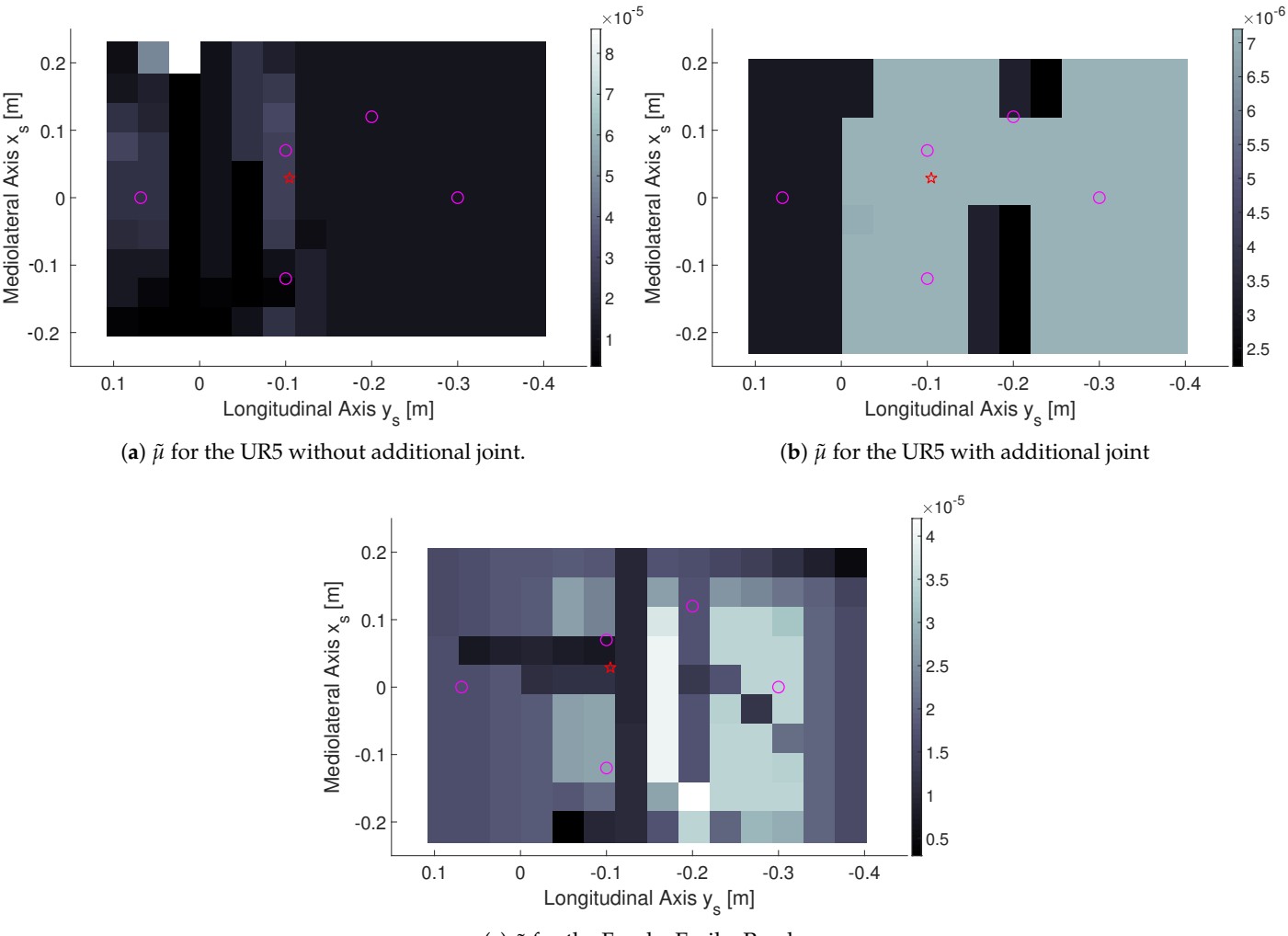

(**a**) $\tilde{\mu}$ for the UR5 without additional joint.

(**b**) $\tilde{\mu}$ for the UR5 with additional joint

(**c**) $\tilde{\mu}$ for the Franka Emika Panda

**Figure 5.** The target metric $\tilde{\mu}$ calculated at the target points for the three arms. The color represents $\mu$ as indicated in the color bars besides the plots. The color scale is different for the three arms. Circles represent points $E_i$, whereas the star represents the heart.

The UR5 with additional joint has a very uniform behavior with a noticeably better overall performance. The combined effects of redundancy and large workspace made the optimization process free of the limitation encountered in the case of the Panda. In the comparison against the UR5 without additional joint, we note that, in the optimization, the angle $\beta$ and the variable $\theta_7$ play a similar role from the point of view of the kinematics. However, $\beta$ is fixed at each optimization step, whereas $\theta_7$ can vary at each target point. Apart from the reported advantage for the UR5 in the proposed optimization, we highlight that the optimality of $\beta$ and therefore of the performance of the arm are limited to these optimization settings. When moving to the real case, it is likely that having a seventh DoF instead of an optimal $\beta$ will increase the performance difference between the two arms.

The decision to introduce an additional, custom degree of freedom improves the efficiency in the accomplishment of the task, but it also introduces several complications that need to be addressed appropriately. First, this joint will have to be actuated, and this will require the design of an actuated joint. It will have to be housed in such a way that the motion is easily transmissible, but does not affect the mechanical performance of the end-effector and the proper execution of the examination. It also becomes necessary to design a special fixing to mount the probe and to leave a sufficient room to allow the correct movement of the probe. Moreover, the addition of an actuated joint complicates the path

towards a successive certification of the device. Despite these complications in the design of the end-effector, in our opinion, the workspace limitations of the Panda arm make it worthwhile to consider adding a seventh DoF to the UR5 robot..

Finally, we make two comments on this study. First, the study is conceived primarily for teleoperation of the robot at the patient site, but the obtained results can be equally applied to a robot at the patient site that carries out USG examination with some degrees of autonomy. Second, the discussion follows uniquely the kinematic optimization, without taking into account practical issues related to the teleoperated control of the arm. These aspects will have to be taken into account in the successive step of designing the whole teleoperated robotic system.

## 6. Conclusions

In this paper, we present the optimization of the design of an end-effector for tele-echocardiography, while comparing a non-redundant solution against two redundant ones. We show that the introduced redundancy allows for a noticeable improvement of manipulability, and that a custom design of an actuated joint to be added to an existing 6 DoFs manipulator is better than adopting a native 7 DoFs Panda arm.

**Author Contributions:** Conceptualization, A.F. and C.A.A.; methodology, A.F. and C.A.A.; software, A.F. and P.G.; validation, A.F. and P.G.; formal analysis, A.F. and P.G.; writing—original draft preparation, A.F. and P.G.; writing—review and editing, C.A.A. and A.F.; supervision, A.F. and C.A.A.; project administration, C.A.A. and A.F.; and funding acquisition, C.A.A. All authors have read and agreed to the published version of the manuscript.

**Funding:** This research received no external funding.

**Institutional Review Board Statement:** Not applicable.

**Informed Consent Statement:** Not applicable.

**Data Availability Statement:** Data available in a publicly accessible repository that does not issue DOIs. This data can be found here: https://www.dropbox.com/s/vl3ch1ndu7bwet7/codeUSGkinopt.rar?dl=0.

**Conflicts of Interest:** The authors declare no conflict of interest.

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
