# Peer review of "Kinematic Optimization for the Design of a Collaborative Robot End-Effector for Tele-Echography†"

_robotics, doi:10.3390/robotics10010008_

Round 1

Reviewer 1 Report

The paper is well-written. The method and result are clearly illustrated and discussed. 

The reviewer has a few remarks:

The literature review could be improved as there are more recent tele-echography robot like Melody from AdEchoTech, Artic from Magellium, FasTele from Japan, etc. There may be other studies involving Franka robot for echography (example: https://www.mdpi.com/2218-6581/9/1/14)

The term collaborative may not be appropriate considering the application of the robot. Although Frank robot does have that functionality, it is not used in this study.

A set of simulations shall be carried out on the three robots to study the evolution of the local criteria at the selected coordinates (from Fig. 2). The average value only does guaranty high performances everywhere. 

The acronym ECG shall be derived in page 4, line 130.

Author Response

Our replies are in the attached document

Reviewer 2 Report

The manuscript is well written and well structured. Theoretical part is adequately explained.

It was not completely clear to me whether the authors assume that the robot performs the task autonomously or whether it is run in manipulator mode by a human operator. Please describe in more detail.

The conclusions section is too brief. More detailed discussion of the results is needed.

Abbreviation COTS in the abstract is not explained.

There are some language issues, in particular mistyping errors that must be corrected.

Reviewer 3 Report

2020-12-12 MDPI Robotics

(1) Regarding the equation 5, the authors said that the value of a weight matrix W_u can be determined by accounting for the task defined in the previous section 3.1. However, I do not understand how the value of each element, w_1..6, can be determined.
(2) In the demonstrated optimization calculation, how the value of each constraint condition, P_sm, P_sM, P_Gm, P_GM, etc. have been given? Since the authors said that "the application of the (task-dependent performance index) method to this problem requires careful tuning of the weights", this part should be explainedmore carefully.
(3) Regarding the process flow explained in Algorithm 1, how the patient's position is updated?
(4) In Table 2 which shows the optimization result, there are two couples of (x_s, y_s, z_s). I guess that the latter one should be (x_G, y_G, z_G).

Round 2

Reviewer 3 Report

I think that the paper has been revised and is well explaining its merit. I do not have any further questions and comments.